# Treatment of Classic Hairy Cell Leukemia: Targeting Minimal Residual Disease beyond Cladribine

**DOI:** 10.3390/cancers14040956

**Published:** 2022-02-15

**Authors:** Jan-Paul Bohn, Sascha Dietrich

**Affiliations:** 1Department of Internal Medicine V, Hematology and Oncology, Medical University of Innsbruck, A-6020 Innsbruck, Austria; 2Department of Medicine V, Hematology, Oncology and Rheumatology, University of Heidelberg, 69120 Heidelberg, Germany; sascha.dietrich@embl.de

**Keywords:** minimal residual disease, rituximab, obinutuzumab, moxetumumab pasudotox, vemurafenib, dabfrafenib, trametinib, ibrutinib

## Abstract

**Simple Summary:**

Standard treatment with purine analogues facilitates a near normal life expectancy in the majority of patients with classic hairy cell leukemia (HCL), a rare chronic B-cell malignancy. However, nearly all patients ultimately relapse and require retreatment, while drug-induced myelotoxicity accumulates predisposing to infectious complications and, possibly, secondary malignancies. Persistence of minimal residual disease (MRD) in a substantial portion of treated patients has become a surrogate for this still limited treatment efficacy. New insights into disease biology initiated design and investigation of several new, chemotherapy-free, targeted drugs with encouraging efficacy in early clinical trials aimed at enhancing eradication of MRD and optimizing drug tolerability. This review provides an update on recent clinical trials investigating treatment strategies beyond purine analogues in HCL and discusses clinically relevant obstacles still to overcome.

**Abstract:**

Classic hairy cell leukemia (HCL) is a rare indolent B-cell lymphoproliferative disorder characterized by profound pancytopenia and frequent infectious complications due to progressive infiltration of the bone marrow and spleen. Lacking effective treatment options, affected patients were confronted with a dismal survival prognosis of less than 5 years when the disease was first described in 1958. Tremendous therapeutic advances were accomplished with the introduction of purine analogues such as cladribine in the 1990s, facilitating a near-normal life expectancy in most HCL patients. Nevertheless, nearly all patients eventually relapse and require successive retreatments, while drug-associated myelotoxicity may accumulate and secondary malignancies may evolve. Detection of minimal residual disease (MRD) in a substantial portion of treated patients has become a surrogate for this still limited treatment efficacy. In the last decade, novel biologic insights such as identification of the driver mutation BRAF V600E have initiated the development and clinical investigation of new, chemotherapy-free, targeted drugs in HCL treatment, with encouraging efficacy in early clinical trials aimed at boosting eradication of MRD while optimizing drug tolerability. This review summarizes current clinical trials investigating treatment strategies beyond purine analogues in HCL and discusses clinically relevant obstacles still to overcome.

## 1. Introduction

Classic hairy cell leukemia (HCL) is a rare indolent B-cell lymphoproliferative disorder characterized by progressive infiltration of the bone marrow and spleen, resulting in pancytopenia and infectious complications [1]. When first described in 1958 by Bouroncle et al. [2], affected patients were confronted with a dismal survival prognosis of less than five years. Only the introduction of interferon-α and, notably, purine analogues (PAs) such as cladribine in the 1990s achieved long-lasting remissions and altered the natural history of the disease [3,4]. At present, HCL patients may face a nearly normal life span compared with the general population [5]. However, nearly all patients eventually relapse after a median follow-up of 15 years, and up to one quarter of them require retreatment within only 5 years [6]. Although re-challenge with PAs is an effective therapeutic strategy in most cases, progression-free survival (PFS) declines with each successive retreatment, while side effects such as myelotoxicity may accumulate and secondary malignancies may evolve [7]. Detection of minimal residual disease (MRD) in a substantial portion of treated patients has become a surrogate for this still limited treatment efficacy. In the last decade, novel insights into disease biology such as identification of the driving gain-of-function mutation BRAF V600E [8] have set new grounds in drug development and started a new impulse of chemotherapy-free, targeted therapeutic approaches in the experimental setting, with promising results aimed at enhancing MRD eradication while reducing drug toxicity [9,10,11,12,13]. This review summarizes current clinical trials investigating treatment strategies beyond purine analogues in HCL and discusses clinically relevant obstacles still to overcome.

## 2. Purine Analogues

For more than 30 years, the PAs cladribine and pentostatin have remained first-line treatments of choice in HCL patients, with overall response rates (ORRs) of >90%, complete remission (CR) rates of >75% and long-lasting PFS durations beyond 20 years for those in deep remission [6]. Cladribine is the most commonly used PA in daily clinical practice, initially administered at 0.1 mg/kg per day by continuous infusion for 7 days in the 1990s [14]. Meanwhile, more convenient and similarly efficacious dosing schedules and routes of administration have been tested, such as a 2 h intravenous infusion or subcutaneous application at 0.14 mg/kg per day for 5 days or weekly for five doses [15,16,17].

The second purine analogue pentostatin is usually administered as a short intravenous infusion at 4 mg/m^2^ followed by hydration every 2 weeks in the outpatient setting [18]. The interrupted dosing schedule every two weeks may be less myelosuppressive than cladribine and permits titration every 3 weeks in case treatment-related higher-grade neutropenia evolves. The advantage of providing a possibly less myelosuppressive and more adaptive dosing schedule in the outpatient setting, however, is counterbalanced by the longer treatment duration generally requiring several months until a response is observed [19]. Although a head-to-head comparative, prospective clinical trial has not been performed, long-term follow-up studies indicate similar clinical efficacies of pentostatin and cladribine in terms of the quality and duration of responses achievable, and both purine analogues can be equally considered front-line treatments of choice in HCL [6]. Although studies with the longest follow-up indeed suggest indefinite remissions in about half of patients in CR and point towards a possibly curative potential of purine analogues in HCL [20], there is a non-diminishing risk of relapse in the long run, particularly for patients achieving partial remission (PR) only. PFS curves do not show a plateau, and patients in PR tend to relapse early within only 5 years [6]. Accordingly, there is a substantial risk of relapse later in life for most patients who are diagnosed at a median age of approximately 55 years [21].

Considering the possibly profound side effects associated with repetitive courses of purine analogues, it thus appears reasonable to break new ground and pursue therapeutic strategies that may further increase PFS and reduce the risk of relapse.

## 3. Minimal Residual Disease

Although HCL cells are highly sensitive to purine analogues, MRD frequently persists in the blood and bone marrow, even in patients with confirmed CR. It has, therefore, been long debated whether patients in CR may actually benefit from MRD monitoring in terms of predicting eventual disease relapse [22]. Sigal and colleagues [23] analyzed 19 HCL patients with confirmed ongoing CR after a median of 16 years from a single course of cladribine and found MRD in 37% (7/19) of cases as assessed by bone marrow immunochemistry; a total of 16% of patients even had morphologic evidence of HCL. On the other hand, Lopez and co-workers [24] reported longer median treatment-free survival in 42 MRD-free HCL patients (not reached) compared to 40 MRD-positive HCL patients (97 months, range: 38–156 months; *p* = 0.05).

With these conflicting data in mind, for now, MRD monitoring in HCL cannot be recommended in clinical practice due to its unclear clinical significance and therapeutic consequence. However, MRD clearance has emerged as an important new endpoint in clinical trials investigating new drugs. After all, the question of whether MRD monitoring is beneficial in terms of PFS and, ultimately, overall survival in a very indolent leukemia with commonly low relapse kinetics such as HCL can only be answered within prospective randomized clinical trials and a longer follow-up.

## 4. Chemotherapy Combined with Anti-CD20 Antibodies

The first intriguing approach to pursue the novel therapeutic goal of MRD negativity was to combine purine analogues with the anti-CD20 monoclonal antibody rituximab, which has been associated with marked improvements in clinical outcomes in combination with chemotherapy in various non-Hodgkin’s lymphomas [25]. CD20, a surface marker of mature B cells, is highly expressed on HCL cells [26], and small phase II studies revealed moderate activity of rituximab as a single agent [27,28,29]. Ravandi et al. [30] conducted the first phase II study with cladribine and rituximab administered as a sequential therapy in eight weekly doses in 59 newly diagnosed HCL patients and reported a CR rate of 100% as well as only one relapse after a median follow-up of 50 months (Table 1). The addition of rituximab was well tolerated without substantial additional toxicity compared with monotherapy [30]. The clinical benefit of combination treatment was confirmed in a smaller cohort of 14 patients in first relapse with a CR rate of 100% and a longer remission duration than with front-line purine analogue treatment in 6 evaluable patients [31]. Following preclinical data suggesting cladribine–rituximab synergy only when both drugs are administered concomitantly due to the short bioavailability of cladribine [32,33], Else and colleagues [33] retrospectively reported 26 mostly multiply relapsed patients receiving four to eight rituximab infusions (375 mg/m^2^) either concurrently (*n* = 20) or sequentially (*n* = 6) with purine analogue treatment (pentostatin, *n* = 15; cladribine = 11). Of 25 patients evaluable for response, there was only 1 non-responder (ORR 96%, CR 88%), and 2 patients relapsed after 56 months (CR) and 10 months (PR); both relapsing patients had received only 4 rituximab infusions (median 6.5 infusions). Notably, PFS after combination therapy was significantly longer than after first-line treatment with purine analogues alone (5-year PFS 87% vs. 35%, and 10-year PFS 87% vs. 12%). Thereby, concomitant treatment was not associated with increased myelotoxicity [33]. Although caution is advised in interpreting these results with respect to limited patient numbers and bias in terms of discrepancies in bone marrow infiltration at diagnosis and relapse as well as in individual patients’ pretreatments, these results certainly illustrate the additional benefit of combination treatment.

To better define the best strategy for combining rituximab with purine analogues, Chihara et al. [34] performed a phase II study randomizing 68 previously untreated HCL patients 1:1 to either cladribine concurrently with eight weekly rituximab infusions (375 mg/m^2^) or cladribine plus delayed rituximab infusions at least 6 months apart in case of MRD positivity according to bone marrow aspirate or blood flow cytometry (Table 1). All patients were allowed to receive another course of rituximab at least 6 months later in case of MRD recurrence. The undetectable MRD rate at 4 weeks as well as the MRD-free CR rate at 6 months was significantly higher in patients receiving concurrent rituximab (62% vs. 9%, and 97% vs. 24% *p* < 0.001). Moreover, the durability of MRD-negative remissions appeared significantly longer with combined treatment (unreached vs. 78.7 months, *p* = 0.001). Thereby, combined treatment did not show substantial differences in cytopenias other than thrombocytopenia without an increase in bleeding and no discrepancy in infections. Of interest, however, another 10 patients initially treated with cladribine only became MRD negative with delayed rituximab infusions after 6 months, resulting in an overall rate of undetectable MRD of 59% (20/34) in this study arm [34]. These efficacy and safety results strongly support a treatment strategy with up-front combination chemoimmunotherapy. Although concurrent administration of cladribine and rituximab appears superior to sequential treatment in terms of eradicating MRD and the duration of MRD-negative disease-free survival, delayed rituximab infusions in MRD-positive patients after cladribine monotherapy may serve as an attractive treatment alternative to achieve the therapeutic goal of MRD negativity with its benefits in long-term PFS, but avoiding additional adverse effects in terms of myelotoxicity and infusion-related reactions.

Other chemotherapeutic backbones being evaluated in combination with rituximab in HCL include bendamustine and pentostatin. Bendamustine incorporates features of both an alkylating agent and a purine analogue and induces apoptotic cell death via multiple mechanisms, including activation of the DNA damage stress response and induction of mitotic catastrophe [35]. A synergistic effect was observed when combined with rituximab in vitro, making the malignant cells more susceptible to bendamustine-associated cytotoxicity [36]. As a result of adequate efficacy and tolerability in several B-cell lymphomas, bendamustine/rituximab is already licensed for first and subsequent lines of treatment in chronic lymphocytic leukemia, follicular lymphoma, marginal zone lymphoma, mantle cell lymphoma and diffuse large B-cell lymphoma [35]. The first evidence of efficacy and safety in HCL was reported in 2013 as part of a tolerability study required prior to recruiting for a larger randomized clinical trial in 12 relapsed and/or refractory HCL patients who received either 70 (*n* = 6) or 90 (*n* = 6) mg/m^2^ bendamustine on days 1 and 2 every 4 weeks for six cycles (Table 1). Rituximab 375 mg/m^2^ was administered on days 1 and 15. With an ORR of 100% and similar CR rates in both dosing cohorts (4 of 6 at 90 mg/m^2^ vs. 3 of 6 at 70 mg/m^2^ bedamustine), rates of undetectable MRD (67% vs. 33%) and time to CR (111 days vs. 223 days) were superior in the higher dosing cohort [37]. As such, 90 mg/m^2^ bendamustine is being administered in a larger long-term clinical trial randomizing multiply relapsed HCL patients to rituximab plus either bendamustine or pentostatin, which is still ongoing (NCT01059786).

## 5. Moxetumumab Pasudotox

Considering the limited efficacy with rituximab monotherapy in HCL, recombinant immunotoxins were designed to improve the quality of responses without implementing additional myelotoxicity typically associated with a chemotherapeutic backbone [38]. Recombinant immunotoxins are composed of the Fv fragment of an antibody connected to a bacterial toxin, which hampers protein synthesis and, thus, induces apoptotic cell death [39]. Targeting CD22 with ‘high affinity’, the disulfide-stabilized recombinant immunotoxin HA22 contains the truncated Pseudomonas exotoxin [40]. Following encouraging efficacy and safety data from phase I testing as early as 2007 [41] that recently found confirmation in a pivotal phase III study [10], the frequently renamed agent HA22→CAT-8015→moxetumumab pasudotox gained approval by the FDA and EMA under the name Lumoxiti for relapsed HCL patients with at least two previous lines of therapy, including one purine analogue regimen. The single-arm registration trial recently reported longer follow-up data of 80 included patients with relapsed and/or refractory HCL who received moxetumumab pasudotox 40 µg/kg intravenously on days 1, 3 and 5 every 4 weeks for a median of six cycles (Table 2). A total of 41% of patients achieved CR, of which 82% proved MRD negative as detected by FACS analysis of bone marrow aspirate and immunohistochemistry analysis of bone marrow tissue biopsy [42]. The primary endpoint of durable CR defined as ongoing hematologic CR for at least 6 months was met in 36% of patients at a median follow-up of 24.6 months. Tagged with a boxed warning by the FDA and EMA, drug-associated hemolytic uremic and capillary leak syndromes occurred in less than 10% of patients and were depicted as generally reversible. However, caution is advised in interpreting this favorable toxicity profile as patients were guaranteed close monitoring during the week of treatment. Expert recommendations suggest adequate oral hydration of approximately 250 mL of water per hour from days 1 to 8 and advise against sleeping more than 2–3 h at a time without drinking. Intravenous crystalline hydration should be restricted to treatment days 1, 3 and 5 in the volume of 1 L before and after moxetumumab pasudotox infusion to avoid fluid overload in terms of pleural effusions and pulmonary edema. In case of emerging drug-related fever, nausea or headache, a short course of dexamethasone 4 mg is recommended to maintain adequate hydration [43]. Of interest, consolidation treatment with 2–4 cycles after confirmed CR appeared to prolong PFS (48.9 vs. 19.1 months) and reduce the risk of relapse (none vs. nine patients). Consequently, 2–4 consolidation cycles are recommended in HCL patients to improve efficacy [43]. Furthermore, prior splenectomy may be predictive of an inferior clinical outcome, with none of eight affected patients achieving CR [44]. Mechanistically, in splenectomized patients, HCL bone marrow infiltration is thought to be more pronounced when severe cytopenias develop, making it, thus, more difficult for moxetumumab pasudotox to penetrate. Importantly, emerging drug-neutralizing antibodies reported in more than one third of patients included in phase I testing were not associated with either additional toxicity or inferior drug pharmacokinetics. Quite the opposite, neutralizing antibodies may be eradicated by repeated drug infusions, and cytotoxic drug levels are maintained. As such, testing for emerging anti-drug antibodies is not recommended [43].

Following these promising efficacy data and the feasible safety profile, moxetumumab pasudotox serves as a valuable, timely limited, chemotherapy-free treatment option in heavily pretreated HCL patients relapsed and/or refractory to standard chemo(immuno)therapy.

To further deepen efficacy in terms of MRD negativity and avoid emerging humeral immunity, a phase II study has been launched to investigate moxetumumab pasudotox in combination with rituximab in 20 relapsed and/or refractory HCL patients (NCT03805932).

On 23 July 2021, the marketing authorization for Lumoxiti (moxetumumab pasudotox) was withdrawn at the request of the marketing authorization holder upon commercial reasons in the European Union. However, Lumoxiti will remain available in the United States (US) and through international pharmacy organizations outside the US.

## 6. Targeting BRAF and MEK

The 2011 discovery of the activating mutation BRAF V600E as the driving genetic event detectable in nearly all HCL patients rapidly added new potential chemotherapy-free players to the therapeutic armamentarium: inhibitors of BRAF and MEK (Figure 1). Shortly thereafter, the first reports documented encouraging efficacy in HCL patients treated with vemurafenib [45], a BRAF inhibitor approved for first-line treatment of unresectable or metastatic melanoma. In the pivotal two European and US studies reported by Tiacci et al. [9] in 2015, overall, 54 BRAF-mutated HCL patients relapsed and/or refractory to purine analogue-based treatments received vemurafenib at the standard melanoma dose of 960 mg b.i.d. for a median of 16 weeks and 18 weeks, respectively (Table 3). The ORR nearly reached 100%, except for one patient presenting with very untypical disease features, and 38% of patients achieved CR. However, MRD, as determined by immunohistochemistry, was detectable in all CR patients, and relapses commonly occurred soon after drug cessation, with the median PFS confined to 9 months in the Italian study [9]. Other than continuous dosing as conducted with clinically aggressive melanoma, vemurafenib treatment was confined to 4 months due to concerns of secondary skin cancers. Indeed, only four patients developed secondary skin cancers (cutaneous basal cell carcinoma, superficial melanoma, squamous cell carcinoma) that were all successfully treated with excision only. Still, low-grade reversible adverse events such as rash and arthralgia/arthritis provoked dose reductions in >50% of patients [9].

Provocatively, in the initial melanoma study, the antitumor efficacy of vemurafenib was demonstrated at a dosage of only 240 mg b.i.d. [46], and a retrospective analysis of 21 HCL patients treated with lower dosing regimens (240–1920 mg/d) outside of clinical trials confirmed dose-independent high response rates by demonstrating switched-off BRAFV600E-induced signaling at doses as low as 240 mg b.i.d. [47]. Furthermore, follow-up data of this series illustrated the efficacy and feasibility of either intermittent dosing with a potential reduction in side effects during drug holidays as well as continuous treatment in patients with very short remissions after drug cessation and a high risk for infection at relapse [48]. Although this accumulating evidence underlines vemurafenib’s efficacy in HCL and reinforces the key pathogenic role of MAPK signaling, remissions induced with vemurafenib seem inferior to those achievable with purine analogues as MRD frequently persists in the bone marrow [9]. In the absence of any additional myelotoxicity, however, vemurafenib may serve as an ideal salvage or bridging treatment for those who may not tolerate the profound side effects associated with chemotherapy, i.e., due to active infections [49].

Considering frequently persisting residual hairy cells even in complete responders, however, relapse appears inevitable with vemurafenib monotherapy after the end of treatment or due to emerging acquired resistance. Mechanisms of resistance have been studied extensively in BRAF-mutant solid tumors such as melanoma or colorectal cancer. In melanoma, acquired resistance to vemurafenib is a common result from MEK-ERK signaling reactivation achieved by diverse genetic mechanisms, including gain-of-function mutations in NRAS and KRAS, mutant BRAF amplification or MAP2K1 mutation [50,51,52]. In an orphan disease such as HCL, the predominant mechanism remains to be better defined, but sequencing studies of individual relapsed patients on treatment with vemurafenib revealed multiple bypassing mutations, including KRAS and MAP2K1, that elicit reactivation of MEK-ERK signaling [53,54]. These clinical data are coherent with recent gene expression analysis showing a sharp silencing of BRAF-MEK-ERK-associated transcriptional output when HCL cells were treated with not only a BRAF but also a MEK inhibitor [55].

Recapitulating this growing clinical and preclinical evidence of acquired resistance strongly favors the combined use of BRAF and MEK inhibitors in HCL treatment. Kreitman and co-workers [11] therefore conducted a phase II study investigating the BRAF inhibitor dabrafenib (150 mg b.i.d.) combined with the MEK inhibitor trametinib (2 mg once daily) for a median duration of 17 months in 43 relapsed and/or refractory HCL patients (Table 3). Although the ORR (78%) and CR rate (49%) did not appear to significantly differ from previous data with a BRAF inhibitor alone [9], CRs with combination treatment were found free of MRD in 30% of cases. Toxicity appeared favorable with only 12% of patients discontinuing treatment, but grade 3–4 adverse events were documented in 49% of cases [11].

Aimed at further reducing treatment-related toxicity with a timely limited, chemotherapy-free therapeutic approach, as well as boosting MRD eradication, Tiacci et al. [12] investigated the efficacy and safety of vemurafenib (960 mg b.i.d. for 8 weeks) in combination with eight biweekly rituximab (375 mg/m^2^) infusions in 30 HCL patients with relapsed and/or refractory disease (Table 3). In total, 26 patients (87%) achieved CR, of which 17 (65%) were free of MRD at the end of treatment, as assessed by PCR testing for BRAF V600E in the bone marrow aspirate and peripheral blood. Early CR confirmation after four weeks of therapy appeared to be predictive of MRD eradication, whereas all four patients in CR relapsing within the observation period failed to show MRD clearance at the end of treatment. Overall, 78% of patients remained free of relapse with a median follow-up of 37 months [12].

Combined treatment did not reveal any additional toxicity, although transient vemurafenib dose reductions due to previously described low-grade adverse events such as rash, photosensitivity and arthralgias were reported in 58% of patients. Remarkably, all patients previously treated with BRAF inhibitors achieved CR, and durations of responses were superior compared to monotherapy. However, relapse-free survival tended to be shorter in patients pretreated with a BRAF inhibitor, suggesting that combination treatment may be preferred to vemurafenib alone in eligible patients [12]. Further supporting combination treatment, the addition of rituximab to vemurafenib appears highly more efficacious compared to phase II data on vemurafenib monotherapy, with CR rates of 87% vs. 35%, and a median time to CR of 4 weeks vs. 8 weeks, respectively [9,12].

Park and colleagues [56] evaluated vemurafenib at a dosage of 960 mg b.i.d. for 16 weeks in combination with obinutuzumab, another anti-CD20 antibody with enhanced cytotoxicity, at a 1000 mg standard dose starting at week 5 in 11 treatment-naïve HCL patients (Table 3). Of nine patients having completed treatment, the ORR was 100%, and the MRD-negative CR rate rose from 7/9 at month 4 to 9/9 at month 10, as evaluated by digital PCR testing for BRAF V600E. All patients remained free of progression with a relatively short median follow-up of only 9.7 months. Vemurafenib-related toxic effects were similar to those reported in previous trials and usually grade 1–2. However, seven patients required dose reductions due to rash (*n* = 5) and arthralgia (*n* = 2) [56].

Overall, these early clinical efficacy data of BRAF inhibitor treatment combined with anti-CD20 blockade compare favorably with purine analogue-based HCL therapy and warrant further validation within a prospective comparative trial.

## 7. Targeting Bruton’s Tyrosine Kinase

B-cell receptor downstream signaling via Bruton’s tyrosine kinase (BTK) has been identified as a driving force in the growth, survival and homing properties of the malignant B cells in several mature B-cell malignancies [57]. The BTK inhibitor ibrutinib is currently approved for CLL [58], mantle cell lymphoma [59] and Waldenstrom macroglobulinemia [60], but its role in HCL is less well defined.

Based on encouraging preclinical data demonstrating decreased MEK-ERK signaling, inhibition of proliferation and reduced survival of HCL cells when exposed to ibrutinib [61], a single-arm phase II study was initiated investigating ibrutinib at a daily dose of either 420 mg (*n* = 24) or 840 mg (*n* = 13) in 37 patients with relapsed HCL or its variant (Table 4) [13]. The primary endpoint of the ORR at 32 weeks was only met in 24% of patients, increasing to a 54% best ORR with a median follow-up of 42 months on treatment. Moreover, responses with continuous treatment seem stable over time, as reflected by an estimated 3-year PFS of 73% and OS of 85%. Including hematologic toxicity, the tolerability profile of ibrutinib in HCL is concordant with previously reported adverse events in other B-cell malignancies and comprises low-grade diarrhea, fatigue, myalgia and nausea in about 50–60% of patients [13]. Considering the favorable PFS and feasible drug tolerability in heavily pretreated HCL patients, ibrutinib may serve as a valuable therapeutic alternative when chemo(immuno)therapy is not an option and/or in BRAF wildtype HCL.

## 8. Targeting B Cell Lymphoma 2

The B cell lymphoma 2 (Bcl-2) inhibitor venetoclax is currently approved for treatment of chronic lymphocytic leukemia and acute myeloid leukemia in adult patients ineligible for intensive chemotherapy. Although HCL cells frequently express Bcl-2, there is no clinical evidence of its efficacy in HCL thus far. However, Vereertbrugghen and colleagues [62] recently investigated its efficacy in isolated primary HCL cells in vitro and demonstrated apoptosis induction with clinically relevant drug concentrations from 0.1 to 1 µM. Further evaluation included co-culture with elements from the local microenvironment, i.e., T lymphocytes and stromal cells, known to benefit HCL cell proliferation and survival, resulting in resistance to venetoclax-induced cell death. These promising preclinical data support the clinical investigation of venetoclax in HCL patients and suggest combinations with drugs targeting the local microenvironment to further improve treatment efficacy.

## 9. Chimeric Antigen Receptor-Engineered (CAR) T-Cell Therapy

Chimeric antigen receptor-engineered (CAR) T-cell therapy has emerged as the most complex immunotherapeutic approach in recent years, primarily directed against CD19+ B-cell malignancies [63]. CARs are genetically engineered synthetic constructs transfected into immunocompetent T cells of the patients to recognize a specific antigen on the surface of tumor cells [63]. Several clinical trials have demonstrated the efficacy of CD19-directed CAR T-cell therapy, leading to FDA and EMA approval in certain B-cell lymphomas [64,65]. Meanwhile, novel CAR constructs are underway targeting other B-cell antigens, such as CD22 [66]. As CD22 is frequently expressed on HCL cells, a phase I study has been launched investigating anti-CD22 CARs in HCL patients with relapsed/refractory disease and/or its variant form (NCT04815356).

## 10. The Remaining Role of Interferon-α in HCL

Interferon-α has been fading in the background of current treatment algorithms after PA introduction with excellent survival outcomes in the majority of patients [6]. Characterized by a favorable toxicity profile compared to PAs in terms of myelotoxicity, however, Interferon-α remains a valid treatment option in patients not qualifying for PAs, i.e., due to infectious complications [67]. With recent data suggesting a possible role in reducing SARS-CoV-2-related inflammation, this may specifically apply in the event of a coronavirus disease 2019 (COVID-19) infection [68]. Moreover, Interferon-α constitutes a valuable therapeutic alternative in pregnant women requiring anti-HCL treatment, when chemotherapy and novel agents may be avoided [69].

## 11. Conclusions

Over the last decade, novel insights into disease biology have initiated the investigation of new targeted agents in the treatment of HCL. Although associated with only modest clinical efficacy when administered as monotherapy, the anti-CD20 antibody rituximab significantly enhances the rate of MRD negativity when administered in combination with cladribine and may, thus, reduce the risk of relapse and decrease the courses of chemotherapeutic retreatment and accompanied cumulative toxicity in individual patients with a longer follow-up. Identifying constitutive BRAF-MEK-ERK signaling as the key oncogenic driver in HCL has implemented inhibitors of BRAF and MEK licensed for melanoma treatment in several clinical trials. With high response rates approaching nearly 100% in relapsed and/or refractory HCL patients, treatment with BRAF and MEK inhibitors underscores the susceptibility of HCL cells when constitutive BRAF signaling is abrogated. Although hairy cells frequently persist in the bone marrow with BRAF inhibitor monotherapy, combination with an anti-CD20 antibody such as rituximab is able to improve both the quality and duration of responses achievable by MRD eradication in most patients. Thereby, the combination of vemurafenib and rituximab may serve as the first chemotherapy-free and timely limited treatment option in HCL, potentially approaching the clinical efficacy of standard treatment with purine analogues ± rituximab with a longer follow-up. Continuous combined inhibition of BRAF and MEK represents another promising chemotherapy-free scheme achieving high rates of MRD and avoiding infusion-related reactions and/or infections possibly associated with CD20 blockade. On the contrary, the BTK inhibitor ibrutinib administered continuously may be a valuable therapeutic option in BRAF-unmutated patients not qualifying for standard treatment. Superior in terms of tolerability when compared to current standard therapy with purine analogues, these novel targeted agents represent a convenient, effective and feasible addition to the treatment spectrum of HCL in the coming years. Longer follow-ups and comparative clinical trials provided, chemotherapy-free agents may ultimately supersede long-established first-line treatment with purine analogues in HCL.

## Figures and Tables

**Figure 1 cancers-14-00956-f001:**
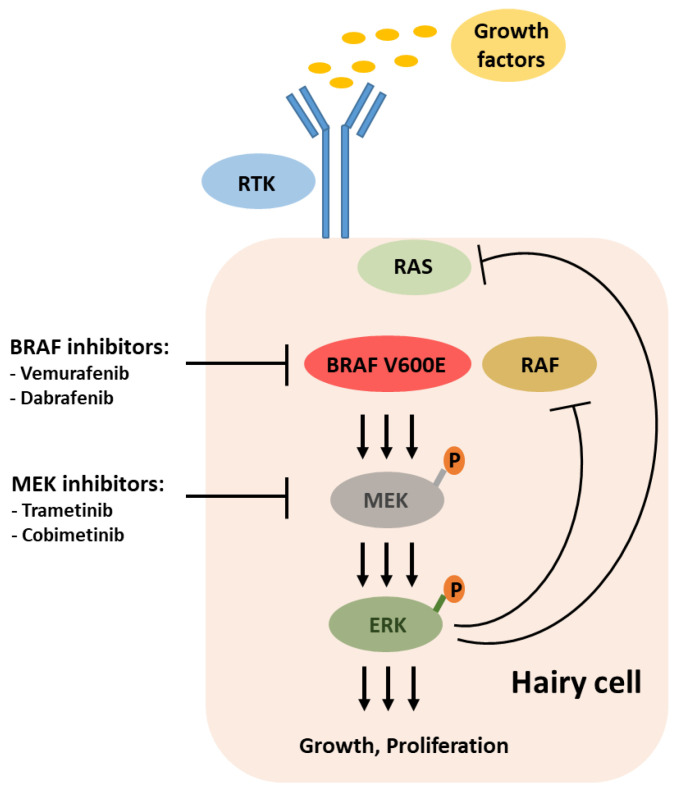
BRAF-MEK-ERK signaling pathway in classic hairy cell leukemia and targets for therapeutic intervention. ERK-dependent negative feedback abrogates RTK-mediated RAS activation, making BRAF V600E a functional monomer.

**Table 1 cancers-14-00956-t001:** Trials to assess the efficacy of chemotherapy + rituximab in patients with classic hairy cell leukemia.

Trial	Phase	Drugs	Disease Status	Patients	ORR (%)	CR (%)	MRD Free (%)	PFS (%)
Ravandi et al. [30]	II	2-CdA + R	Untreated Relapsed	59 14	100 100	100 100	76 64	95 (5-year) 100 (5-year)
Chihara et al. [34]	II	2-CdA + R vs. delayed	Relapsed	68	100	100 vs. 88	97 vs. 24	94 vs. 12
Burotto et al. [37]	II	Benda + R	Relapsed	12	100	50 vs. 67	67 vs. 100	31 months for patients in CR

ORR, overall response rate; CR, complete remission; MRD, minimal residual disease; PFS, progression-free survival; 2-CdA, cladribine; R, rituximab; Benda, bendamustine.

**Table 2 cancers-14-00956-t002:** Trials to assess the efficacy of moxetumumab pasudotox in patients with classic hairy cell leukemia.

Trial	Phase	Drugs	Disease Status	Patients	ORR (%)	CR (%)	MRD Free (%)
Kreitman et al. [41]	I	Moxe (5–40 µg/kg) Moxe (50 µg/kg)	Relapsed Relapsed	16 33	86 88	57 64	n.a. 33
Kreitman et al. [10,42]	III	Moxe (40 µg/kg)	Relapsed	80	75	41	34
NCT03805932	I	Moxe (40 µg/kg) + R	Relapsed	20	ongoing	ongoing	ongoing

ORR, overall response rate; CR, complete remission; MRD minimal residual disease; Moxe, moxetumumab pasudotox; R, rituximab.

**Table 3 cancers-14-00956-t003:** Trials to assess the efficacy of inhibitors of BRAF and MEK in patients with classic hairy cell leukemia.

Trial	Phase	Drugs	Disease Status	Patients	ORR (%)	CR (%)	MRD Free (%)	1-Year PFS (%)
Tiacci et al. [9]	II	Vem	Relapsed	54	100	38	0	73
Tiacci et al. [12]	II	Vem + R	Relapsed	30	100	87	65	78 (3-year)
Kreitman et al. [11]	II	Dabra + Tram	Relapsed	43	78	49	15	98
Park et al. [56]	I	Vem + Obi	Untreated	9	100	100	100	9.7 months PFS

ORR, overall response rate; CR, complete remission; MRD, minimal residual disease; PFS, progression-free survival; Vem, vemurafenib; R, rituximab; Dabra, dabrafenib; Tram, trametinib; Obi, obinutuzumab.

**Table 4 cancers-14-00956-t004:** Trials to assess the efficacy of Bruton’s tyrosine kinase inhibitors in patients with classic hairy cell leukemia.

Trial	Phase	Drugs	Disease Status	Patients	ORR (%)	CR (%)	MRD Free (%)	3-Year PFS (%)
Rogers et al. [13]	II	Ibrutinib	Relapsed	37	73	0	0	73

ORR, overall response rate; CR, complete remission; MRD, minimal residual disease; PFS, progression-free survival.

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
