# Peer review of "Treatment of Classic Hairy Cell Leukemia: Targeting Minimal Residual Disease beyond Cladribine"

_cancers, 2022, doi:10.3390/cancers14040956_

Round 1

Reviewer 1 Report

The two authors have written an excellent comprehensive review on the treatment of cHCL. I only have minor comments.

  1. Although I am not convinced that MRD should be the surrogate endpoint in cHCL, the authors discuss this appropriately. But can the authors make a statement? Should we measure MRD? If so, how? nested PCR, flow or ddPCR? If not, why should we aim for combination therapies
  2. Is there any role left for interferon?

Any future perspectives?

  1. Although the majority of novel treatments target the BRAF-ERK pathway, are there any treatment opportunities based on the mutation profile of cHL (for example the presence of KMT2C and CDKN1
  2. Any room for BH3 mimetics?
    Frontiers | In Vitro Sensitivity to Venetoclax and Microenvironment Protection in Hairy Cell Leukemia | Oncology (frontiersin.org)
  3. CART?
    Phase I Study of Anti-CD22 Chimeric Receptor T Cells in Patients With Relapsed/Refractory Hairy Cell Leukemia and Variant - Full Text View - ClinicalTrials.gov
  4. Bispecifics?

Reviewer 2 Report

In this review, authors summarize current clinical treatments (trials) for HCL beyond Cladribine-containing regimen. I think some changes are needed to make the work more valuable.

  1. I would suggest to add a list of acronyms used in the paper at the beginning of the review, before Introduction section.
  2. Since the issue of MRD is addressed in all the paragraphs of the paper, I would add a reference to MRD in the title (i.e.: treatment of classic hairy cell leukemia: MRD significance beyond Cladribine… or something similar).
  3. For the same reason I would put the paragraph 3 on MRD immediately after the Introduction section.
  4. Please, format tables better. Do not break the line with the words in the same box. I would divide Table I into four distinct tables, each within its own paragraph. Add a legend to table/s with abbreviations used in the table/s.
  5. Add a reference/s at line 314 about Ibrutinib in CLL, MCL, and Waldenstrom.
  6. Correct typos.

Round 2

Reviewer 2 Report

I am satisfied with the improvements made and I think the work is suitable for publication in Cancers.

Please correct Akronyms in Acronyms (line 26)